# Perceptions of Patient Safety Culture among Triage Nurses in the Emergency Department: A Cross-Sectional Study

**DOI:** 10.3390/healthcare11243155

**Published:** 2023-12-12

**Authors:** Zvonka Fekonja, Sergej Kmetec, Nataša Mlinar Reljić, Jožica Černe Kolarič, Majda Pajnkihar, Matej Strnad

**Affiliations:** 1Faculty of Health Sciences, University of Maribor, 2000 Maribor, Slovenia; sergej.kmetec1@um.si (S.K.); natasa.mlinar@um.si (N.M.R.); jozica.cerne1@um.si (J.Č.K.); majda.pajnkihar@um.si (M.P.); 2Faculty of Medicine, University of Maribor, 2000 Maribor, Slovenia; matej.strnad@um.si; 3Emergency Department, University Clinical Centre Maribor, 2000 Maribor, Slovenia; 4Center for Emergency Medicine, Prehospital Unit, Community Healthcare Center, 2000 Maribor, Slovenia

**Keywords:** triage, patient safety, perception, safety management, Emergency Medical Services

## Abstract

The patient safety culture is key to ensuring patient safety in healthcare organizations. The triage environment is inherently demanding for patient safety and is characterized by high stress, rapid decision-making, and quick action. In several countries, including Slovenia, there is a lack of studies on the patient safety culture among triage nurses. This study aimed to assess the perceptions of the patient safety culture among triage nurses. A cross-sectional survey design was used. The Emergency Medical Services-Safety Attitudes Questionnaire, distributed to triage nurses, was used to collect data. A total of 201 triage nurses participated in this study. The results revealed that the overall average perception of the patient safety culture was 57.27% (*SD* = 57.27), indicating that the perception of the patient safety culture among triage nurses in the emergency department was non-positive and requires improvement. “Job Satisfaction” received the highest score (63.18%; *SD* = 17.19), while “Working Conditions” received the lowest (49.91%; *SD* = 17.37). The perception of positive and negative safety culture responses was statistically significant for age (*χ*^2^ (3) = 17.750, *p* ≤ 0.001), education (*χ*^2^ (2) = 6.957, *p* = 0.031) and length of working experience (*χ*^2^ (3) = 8.875, *p* = 0.031). The findings emphasize the significance of improving the safety culture in relation to several areas of patient care during the triage process. This research serves as a crucial foundation for enhancing patient safety in triage, providing quality care, and reducing adverse events.

## 1. Introduction

The number of patients in emergency departments is increasing worldwide and often exceeds the capacity to immediately deal with all the needed care. Therefore, in the context of the efficient allocation of limited resources during overcrowding, triage has become a commonly used working tool in the emergency department [1]. Triage, the process of rapidly and accurately assessing a patient’s condition, determining treatment priorities, and allocating healthcare resources, represents the initial and critical interaction between patients and triage nurses [2,3,4]. Hence, the triage nurse must efficiently and accurately evaluate the patient’s health status, determine who should be treated first, and allocate healthcare resources according to priority [5]. The expert assessment of a patient’s urgency is essential in triage to ensure patient safety [6], as it can provide timely care and reduce the risk of poor health prognosis due to prolonged waiting for treatment [2,7,8]. Several factors within the realm of emergency care significantly impact the efficacy of the triage process. These factors encompass resource management and patient flow, patient satisfaction, the expertise and acumen of triage nurses, and the safety culture within the organisation [9].

The safety culture in the emergency department is a part of the organisational culture, encompassing the integration of safety thinking and practices into clinical activities [10]. It is defined as a product of the values, attitudes, perceptions, competencies, and behavioural patterns of individuals and groups that determine the commitment, style, and professionalism in managing health and safety within the organisation [11,12] to make patient safety the highest priority [13] and responsibility [14] of all employees, as patient safety is not the sole responsibility of the individual triage nurse but is a collective endeavour [15]. Furthermore, patient safety relates to avoiding, preventing, and managing adverse events or injuries associated with healthcare processes [16]. Nurses, who are at the forefront of emergency department care, have a particularly significant responsibility in ensuring patient safety and enhancing the culture of patient safety by vigilantly monitoring patients’ clinical conditions, detecting errors, understanding care processes, and addressing system deficiencies to deliver high-quality care [17,18]. Therefore, they must perform nursing activities in a patient-safe manner and strive for activities that reinforce such activities [16].

The safety culture in the emergency department is one of the key priorities that should be considered by managers, healthcare professionals, and decision-makers, as highlighted by Alshyyab et al. [19]. In this regard, regular safety culture measurements are required to provide hospital and emergency department managers with valuable insights into areas that require improvement and assess the quality enhancement plan implemented within healthcare. The Emergency Medical Services-Safety Attitudes Questionnaire (EMS-SAQ) assesses the safety culture in prehospital and emergency healthcare facilities, although it remains an underutilised tool in prehospital facilities, but not in emergency healthcare facilities [20]. Assessing the patient safety culture and implementing impact measures have become necessary [21] in emergency care. Nevertheless, there is scant evidence in this area. Triage nurses, who are responsible for initial assessment and patient prioritisation [7], play a central role in shaping the safety culture [18] in emergency departments, characterised by their inherently fast-paced, physically hazardous, and intensely stressful nature, demanding rapid decision-making and immediate responses. Although patient safety remains the foremost priority and responsibility of all healthcare professionals [14], this study focuses on the perceptions of triage nurses working in the complex and dynamic environment of the emergency department. Understanding their perception of the safety culture is crucial since these perceptions can significantly impact the quality of patient care and safety outcomes. Considering the evidence presented above, the primary objective of this study was to examine how triage nurses perceive the patient safety culture in emergency departments. By exploring their perceptions of the safety culture, we also aimed to gain a comprehensive understanding of how triage nurses view and engage with safety practices in their clinical setting. By focusing on this unique context, the study intends to add valued insights into the perspectives of triage nurses, contributing valuable information to the existing body of literature on advanced patient safety practices in the emergency department.

## 2. Materials and Methods

### 2.1. Design

The study used a quantitative cross-sectional method, utilising a valid survey instrument, to collect data from triage nurses working in the emergency departments in Slovenia. To ensure the reliability of the reporting in our cross-sectional study, we followed the STROBE (Strengthening the Reporting of Observational Studies in Epidemiology) checklist.

### 2.2. Setting and Sample

The study was conducted across all 11 emergency departments operating within regional hospitals or University clinical centres in Slovenia. Nine are operating under general hospitals nationwide, and two are University clinical centres in the two largest cities in Slovenia. According to the latest statistics, 730,445 patients visit the emergency department across the country annually [21]. Convenience sampling was used, characterised by inviting the entire population that is accessible within a certain period and meets the requirements of the inclusion criteria [22]. In Slovenia, triage is performed according to the Manchester Triage System (MTS) principles, which have proven to be the most suitable for implementation in our environment. Moreover, triage is performed by a registered nurse who has acquired additional specialised knowledge in this area and possesses a minimum of three years of experience in the emergency field. According to Slovenian legislation, triage can also be performed by experienced (with experience working in an emergency department) nurse assistants who have also acquired specialised knowledge through the Slovenian Nursing Chamber and based on an employer’s assessment that they are suitable to undertake triage duties [23].

The inclusion criteria were: (1) nurses, including nursing assistants, registered nurses, and those with a master’s degree in nursing, (2) actively engaged in triage duties within the emergency department, and (3) possessing a minimum of one year of triage experience during the (4) designated data collection period. The study excluded nurses with less than one year of triage experience and other healthcare professionals (who do not belong to the nursing profession) in the emergency department not involved in the triage process.

In accordance with the Cochran formula [24] and considering the number of triage nurses employed in 2020 (*n* = 230, recognising that this number is subject to continuous fluctuations due to staff turnover), our estimation indicated that the required sample size should be 148 triage nurses. Out of the 240 questionnaires distributed, 206 were returned, resulting in a response rate of 86%. Subsequently, five questionnaires were excluded due to a non-completion rate of 50%, ultimately yielding a final sample size of 201 participants.

### 2.3. Study Tools/Instruments

The data were collected using the Emergency Medical Services-Safety Attitudes Questionnaire (EMS-SAQ) validated tool. Permission to utilise the EMS-SAQ was obtained after consultation with the instrument’s author. The EMS-SAQ consists of 30 core items, which collectively encompass six domains related to the safety culture: (1) Safety climate (seven items), (2) Job satisfaction (five items), (3) Perception of management (four items), (4) Teamwork climate (six items), (5) Working conditions (four items), and (6) Stress recognition (four items). Respondents provided their answers to each question using a five-point Likert scale, ranging from 1 (“strongly disagree”) to 5 (“strongly agree”). Demographic data of the participants, such as age, gender, job titles, level of education, work shifts, and education and training, were obtained using a descriptive data tool developed by the researchers.

### 2.4. Data Collection

Data were collected between May 2020 and April 2021. The first author (Z.F.) personally distributed the questionnaires to all eligible participants and informed them about the study’s objectives. Upon voluntary agreement to participate, the head nurses were entrusted with envelopes containing the questionnaires, which they then distributed to the eligible triage nurses. Subsequently, participants received an envelope with a copy of the EMS-SAQ questionnaire and a pen to complete the survey instrument. They were asked to complete the instrument at a convenient time and return it to the researcher in a sealed envelope at a pre-determined date and time within seven days at an agreed location.

### 2.5. Data Analysis

After collection, the data were entered into SPSS Statistic 25 software. The statistical analysis in this study involved both descriptive and inferential statistics. Descriptive statistics were applied for the demographic data, including frequency, percentages, mean, and standard deviation calculations. Regarding statements concerning attitudes towards the patient safety culture, a 5-point Likert scale was utilised, with ratings ranging from “strongly disagree” (1) to “strongly agree” (5). This Likert scale was converted into a 100-point scale, with score values of 0 (“strongly disagree”), 25 (“slightly disagree”), 50 (“neutral”), 75 (“slightly agree”), and 100 (“strongly agree”). Three items (2, 11, 36) which were negatively worded were reversed and recoded in the analysis to match the other items. Results were considered positive if their total value equalled or exceeded 75, corresponding to “somewhat agree” on the original 5-point Likert scale. This indicated a positive perception of the safety culture among the participants [25,26,27]. Therefore, on average, respondents had to answer “somewhat agree” or more to be categorised as having a positive perception of safety culture. The significance of differences between the mean values in the compared groups was analysed by following the rules of statistical test selection. The dispersion of the variables was evaluated using the Shapiro–Wilk test, which revealed that the data did not follow a normal distribution.

Consequently, the significance of differences between groups was examined using either the Kruskal–Wallis or Mann–Whitney U-test. All analyses used a significance level of less than 0.05 to determine statistical significance. Additionally, Cronbach’s α coefficient was calculated to evaluate the internal consistency and reliability of the EMS-SAQ instrument.

### 2.6. Ethical Considerations

Prior to conducting the survey, approval from the Medical Ethics Committee of the Republic of Slovenia (Ref. No: 0120-558/2017/14) was obtained. Permission to use the instruments was obtained directly from the authors. Consent was also obtained from the management of all participating emergency departments. In addition, as the study does not pose any risk or hazard to the participants, we opted for informed verbal consent, which was obtained from triage nurses after giving them a participant information sheet, which was attached to the first page of the questionnaire, stating the aim and all necessary information about the study. Confidentiality, anonymity, and voluntary withdrawal from participation before submitting the questionnaire were emphasised.

## 3. Results

The survey involved 201 emergency department triage nurses working in 11 emergency departments across Slovenia. The Cronbach’s α coefficient of the EMS-SAQ instrument was 0.81, indicating good reliability. The demographics of the survey participants are detailed in Table 1.

The study included 42 (21%) male and 159 (79%) female participants. Among the 201 triage nurses, 139 (69%) were registered nurses, 42 (21%) were nursing assistants, and 20 (10%) held a master’s degree in nursing. The average age of the triage nurses was 37 years (*SD* = 9.89), with the youngest being 22 and the oldest 61. The average length of work experience was 14 years (*SD* = 10.34), of which 65 (32%) had more than 16 years of experience, followed by those with less than five years of experience (*n* = 53; 26%), 45 (22%) having 11–15 years of experience, and 38 (19%) with 6–10 years of experience. The average length of work experience in the emergency department was nine years (*SD* = 8.40). The most experienced triage nurse had 36 years of experience in the emergency department and 39 years of total work experience.

Triage nurses perform triage based on the principles of the Manchester Triage System for an average of 5 years (*SD* = 3.45), during which they encounter an average of 63 patients per day (*SD* = 40.81). Most triage nurses primarily work in a shift that lasts from 7:00 to 19:00 (*n* = 91, 45%), followed by those working in the shift from 15:00 to 22:00 (*n* = 39, 19%). The triage workday for triage nurses is mostly 8 h long (*n* = 104, 52%), followed by a 12 h shift (*n* = 95, 47%), with two (1%) triage nurses performing triage for 4 h. Nearly half of triage nurses (*n* = 100; 50%) worked in triage for more than four consecutive days, while 29 (14%) worked continuously for three to four days.

The average total score of the EMS-SAQ was 57.27 (*SD* = 13.81) (Table 2). The domain Working conditions had the lowest average rating (*M* = 49.91; *SD* = 17.37). Conversely, job satisfaction had the highest average rating (*M* = 63.18; *SD* = 17.19).

The item with the highest average score was: The emergency department deals with mass disasters that occur (*M* = 75.87; *SD* = 21.57), followed by the item: I have colleagues who will leave the emergency department for employment elsewhere (*M* = 73.76; *SD* = 26.43). This was followed by the item regarding job satisfaction: I like my job (*M* = 72.26; *SD* = 25.35). Other items rated above 70% were: The work organization could do more to improve safety (*M* = 71.64; *SD* = 24.33) and: A confidential reporting system is useful for improving patient safety (*M* = 70.40; *SD* = 24.63). The item with the lowest average score pertained to working conditions: Management constructively deals with problematic staff (*M* = 41.54; *SD* = 21.29). The lowest scores were also given to item 49: Adverse events (incidents in which a patient is harmed due to healthcare or medical equipment failure) occur in the emergency department (*M* = 34.70; *SD* = 22.34), item 48: Accidents frequently happen when dealing with patients (stretchers, beds, patient falls) (*M* = 29.23; *SD* = 24.13), and item 33: I have made a mistake that could potentially harm a patient (*M* = 28.11; *SD* = 23.58).

Figure 1 shows that the domain Stress recognition had the highest positive responses at 40%, followed by Teamwork climate at 28% and Job satisfaction at 26%. Safety climate received the lowest percentage of positive responses, with only 13% of participants rating it positively (Figure 1).

Assessing positive and negative safety culture responses reveals distinct patterns across demographic variables. Regarding gender, most females, constituting 86%, obtained scores below 75, while 14% exceeded this threshold. A comparable trend is observed among males, with 88% scoring below 75 and 12% surpassing 75. Overall, between genders, there is no statistically significant difference. Regarding age as a determinant, participants aged 30 or below exhibited a distribution where 28% scored below 75, while 32% scored above this threshold. Within the 31–40 age group, 47% fell below 75, and 20% exceeded 75. Notably, the 41–50 age cohort demonstrated 19% below 75 and 16% above 75. In the 51 and above age category, 7% scored below 75, while 32% scored above. The chi-square test (*χ*^2^ = 17.750, *p* < 0.001) underscores the statistical significance of associations between age groups and safety culture perceptions. A discernible trend emerges regarding education, with 98% of participants with a vocational nursing education recording scores below 75 and a mere 2% surpassing this threshold. Similarly, 86% fell below 75 for Bachelor of Science in nursing, while 14% exceeded it. In the Master of Science in nursing/master’s in nursing cohort, 75% registered scores below 75, and 25% scored above. The chi-square test (*χ*^2^ = 6.957, *p* = 0.031) substantiates a statistically significant correlation between education levels and safety culture perceptions. Data analysis in the context of working experience reveals a consistent pattern. Among those with ≤5 years of experience, 85% scored below 75, whereas 15% exceeded this threshold. In the 6–10 years category, 87% scored below 75, and 13% scored above. Strikingly, the 11–15 years group indicated uniformity, with 100% scoring below 75 and none surpassing 75. Participants with ≥16 years of experience demonstrated 82% below 75 and 18% above. The chi-square test (*χ*^2^ = 8.875, *p* = 0.031) underscores a statistically significant association between working experience and safety culture perceptions, further emphasising the nuanced nature of these perceptions across demographic variables (Table 3).

## 4. Discussion

This study aimed to assess the perception of the patient safety culture among triage nurses in emergency departments. The study’s findings reveal that the perceptions of triage nurses were negative across all analysed variables, underscoring the necessity for improvements in this domain. The aspect requiring the most significant improvement was Working conditions. Conversely, Job satisfaction had the highest average rating. The perception of positive and negative safety culture responses was statistically significant for age, education, and length of working experience. The domain with the largest percentage of positive responses was Stress recognition, followed by Job satisfaction and Teamwork climate. These findings highlight the need for improving safety culture across various dimensions of patient care within the triage process within the studied emergency departments.

As indicated by AL-Mugheed et al. [28], a negative attitude inhibits the implementation of measures and the enhancement of care in healthcare facilities, impacting patient safety maintenance.

Assessing the level of safety culture is the starting point for improving safety culture within any organisation. Moving towards safer care without a clear understanding of the current state can lead to increased costs and exposure to new risks [29]. The overall average safety culture score was 57 (*SD* = 13.81), indicating triage nurses’ negative perception of the safety culture. This score is higher than in previous studies conducted in Jordan [30], Egypt [31,32], Yemen [33], Saudi Arabia [34], Cyprus [27], Iran [35], while lower than the study from Turkey [36], Brazil [37], the USA [38], Finland [39], and the nine studied countries (Italy, Portugal, the United Kingdom, China, The Netherlands, Germany, France, Finland, and Japan) in [20]. These varying results may be associated with cultural differences, contextual conditions, sampling characteristics, and the instruments used.

In our study, the domain Job Satisfaction obtained the highest score (*M* = 63.18; *SD* = 17.19), and within this domain, the statements that triage nurses enjoy their work and are proud of their work achieve one of the highest scores. These findings suggest that the respondents are engaged in work they are passionate about and find fulfilling. Hence, this outcome should be interpreted as positive, given that the job satisfaction of triage nurses is closely interconnected with the provision of high-quality care. In institutions where professional staff are dissatisfied with their work experience, the likelihood of adverse events is elevated, and staff turnover rates are also higher [40]. Regarding staff turnover, the respondents attributed the second-highest score to the statement that colleagues will leave their positions for employment elsewhere. Sasso et al. [41] indicated that substantial staff turnover could be attributed to inadequate compensation, job insecurity, understaffing leading to excessive workloads, involvement in non-nursing tasks, absence of career development opportunities, and insufficient leadership support.

Furthermore, employees are considered predictors of patient safety; therefore, having a motivated workforce, one of the biggest challenges in hospitals, is crucial [29]. Penconek et al. [42] emphasise that nurse leaders in healthcare organisations play a central role in creating positive working conditions, promoting job satisfaction, and reducing nurse turnover. Respondents believe that the work organisation could do more to enhance safety. According to Boamah et al. [43], improving patient safety in healthcare organisations requires effective leadership at all levels. Enhancing patient safety necessitates institutions embrace a just culture where staff feel comfortable reporting adverse events and discussing them openly [44]. Many researchers argue that leaders’ perceptions are critical to patient safety [37]. Hospital leaders who focus on the patient safety culture must create an environment of communication based on mutual trust, good information flow, a shared understanding of the importance of safety, management, and leadership, and a non-punitive approach to incident and error reporting. Ensuring patient safety must be a central aspect of hospital organisational culture [45,46,47].

Working conditions, particularly within emergency departments, prone to overcrowding and adverse events, influence the perception of the safety culture [37]. The domain of Working Conditions, which relates to the participant’s perception of the quality of their working environment, was rated the lowest among all domains of safety culture perception, indicating a negative attitude toward staffing levels, the training of new staff, the availability of necessary decision-making information, and supervision of new hires. A likely explanation is that respondents are dissatisfied with the workforce and human resources. Our findings align with the results of other studies that demonstrate how working conditions, such as staff turnover, shift work, fatigue, a lack of control over complex and hazardous work conditions, high workloads, crowding, and inadequate workspace, impact patient safety [48]. In addition, staff shortages, increased patient volumes, and higher expectations from other healthcare professionals may contribute to the increased workload, potentially jeopardising patient safety [49]. Studies have shown that improved working conditions can lead to training and monitoring, an adequate workforce, and the maintenance of therapeutic and diagnostic information [50,51].

The percentages of respondents with positive responses (score > 75) for all domains of the safety culture in this study (range 13% to 40% positive) were comparable to studies in emergency departments, intensive care units, and hospitals [25,52,53]. We found that the Recognition of Stress domain received the highest number of positive responses, followed by the Teamwork Climate and Job Satisfaction domains. The Safety Climate domain received the fewest positive responses. In most emergency department studies, the best-rated domains are teamwork, perception of management, and patient safety promotion [10,54,55,56,57]. Predictive factors of a positive safety culture in healthcare organisations, according to the findings of Tear et al. [58], include open communication, effective flow of communication, shared perception of safety communication, organisational learning, commitment of top leadership, and non-punitive approaches to reporting incidents and errors. Awareness of safety, error reporting, gender, and work experience also predict a positive safety culture [59]. Hu et al. [60] emphasise that hospitals should build a patient safety culture guided by the right values and encourage transforming the patient safety culture into a more satisfactory state of dynamic equilibrium. Improving the safety culture in the triage process is a key element in preventing/reducing errors and enhancing the overall quality of healthcare and patient safety [17].

Perceptions of the safety culture varied according to the demographic characteristics of triage nurses. Among the individual characteristics of triage nurses, age, level of education, and total years of working experience were associated with significant differences in perceived safety culture scores. Regarding age, respondents aged 50 and above consistently assigned higher ratings to all aspects of the perceived safety culture compared to those under 50. Similarly Isa et al. [61] and Bondevik et al. [62] identified disparities across age groups influencing adherence to an organisation’s safety culture. Conversely, younger employees benefiting from recent and extensive training in patient safety culture may demonstrate a heightened ability to recognise potential risks. As a result, organisational leadership is urged to acknowledge these generational distinctions and formulate targeted strategies for fostering safety awareness in the workplace. Overall, highly educated triage healthcare personnel in the emergency department rated all safety culture perception domains (five *p* ≤ 0.001) higher than less educated triage nurses. These findings align with previous studies conducted using the EMS-SAQ instrument [38,63] and in contrast to a study conducted in Finland [39]. We found that triage nurses with a master’s degree perceive the safety culture more positively than those with a bachelor’s and high school education. This result aligns with the study by Alquwez, Cruz, Almoghairi, Al-Otaibi, Almutairi, Alicante, and Colet [34] and Malak et al. [64], which reported that nurses with higher education and positions perceived the safety culture more positively than others. This finding can be explained by the fact that those with higher education have more knowledge of policies and procedures related to a positive safety culture gained during their educational programs and ongoing professional development [65] and have better abilities to critically assess all factors influencing the safety culture in the emergency department [39]. Triage nurses with over 16 years of experience in the emergency department rated all safety domains higher than those with less than 15 years of experience. Therefore, as experience increases, so does the perception of patient safety [66,67]. According to Abdul Rahman et al. [68], inexperienced and freshly graduated nurses are likely to experience practice-related stress, which exposes them to higher errors, affecting their perception of patient safety. Many staff with insufficient experience and inadequate attention to patient safety represent a risk to patients and other nurses [69]. Nurses play a central role in improving the patient safety culture [18] and addressing patient safety issues [70] as they monitor patients’ clinical conditions, detect errors, and understand care processes and system weaknesses to provide high-quality care [17]. In work environments with a patient safety-focused program, the leadership can positively influence the effectiveness of employees’ work, their awareness, and their involvement in improving the quality and safety of care [71].

### 4.1. Implications for Nursing Practice

This study presents initial findings regarding triage nurses’ perception and comprehension of the patient safety culture. It reveals that triage nurses in emergency departments have a negative perception of the safety culture, which calls for immediate attention to improve various aspects of patient care in the triage process. It stands as one of the pioneering studies examining the perceptions of triage nurses regarding the safety culture. This study addresses the existing knowledge gap related to the elements that triage nurses consider crucial in establishing a culture of safety in an emergency department and adds the body of knowledge about ensuring patient safety during the triage process. It also serves as a foundation for generating hypotheses and encouraging further investigation. Enhancing the safety culture should be a top priority for healthcare organisations to mitigate errors and improve the overall quality and safety of patient care.

### 4.2. Limitations of the Study

Despite significant findings, several limitations need to be acknowledged. The study’s sample consisted of triage nurses working in Slovenian emergency departments, specifically those involved in the triage process. Consequently, the results may not be readily generalisable to all emergency department nurses internationally. The data collection occurred during the spring and summer months, coinciding with vacation periods, potentially leading to increased workloads and overtime, which might have influenced the study’s outcomes. Data were collected during the COVID-19 pandemic period. The pandemic introduced unique challenges and circumstances that could impact assessing safety culture in the emergency department. The limitations and effects on the data collection process during the COVID-19 pandemic may include disruptions in regular workflows, increased workload and stress among healthcare professionals, changes in protocols and procedures, and potential shifts in the focus of healthcare priorities. These factors could influence the responses of participants and the overall dynamics of the emergency department, potentially impacting the nuanced understanding of the safety culture. Another limitation of this study is the chance that participants may provide socially desirable responses, which is inherent to the data collection process. The Emergency Medical Services-Safety Attitudes Questionnaire (EMS-SAQ) evaluates the safety culture within prehospital and emergency healthcare settings. Despite its efficacy in emergency healthcare, the EMS-SAQ remains underutilized in prehospital settings. This underutilization in prehospital settings presents a limitation in the broader application of the EMS-SAQ, as it hinders the comprehensive assessment of the safety culture across the entire continuum of emergency care.

Furthermore, the study used a cross-sectional design that did not identify or predict cause and effect between the independent and dependent variables of the study. Another limitation is the scarcity of research on measuring the safety culture among triage nurses, limiting the ability to compare this study’s findings with other national and international studies; therefore, the current study should be extended to a wider sample of triage nurses abroad. Despite its limitations, this study represents one important dimension of the prevailing safety culture in Slovenian emergency departments and settings with similar characteristics, but further research is needed.

## 5. Conclusions

This study provides some initial evidence on triage nurses’ perceptions of the patient safety culture and adds to the body of evidence about which factors are significant to triage nurses in creating a culture of safety in the emergency department. Triage nurses in our study perceived the safety climate negatively across all patient safety domains, contributing to identifying areas needing improvement. The domain of Job satisfaction received the highest average perception rating while Working conditions received the lowest perception rating among triage nurses. Concerning experience, participants with fewer years of experience showed lower average perceptions of patient safety attitudes than their more experienced counterparts. Regular assessments of the patient safety culture in emergency departments are necessary to provide hospital leadership with essential information about the areas of the safety culture that require improvement and to evaluate implemented plans to enhance patient safety and reduce adverse events. The findings delineated in this study emphasize the imperative for precision-targeted interventions and strategic initiatives designed to cultivate a robust safety culture. This is integral to augmenting the holistic well-being of healthcare professionals, especially triage nurses and patients. Through the conscientious recognition and comprehensive resolution of identified issues, we lay the groundwork for a transformative paradigm shift, ensuring an optimally safe and conducive work milieu for triage nurses.

## Figures and Tables

**Figure 1 healthcare-11-03155-f001:**
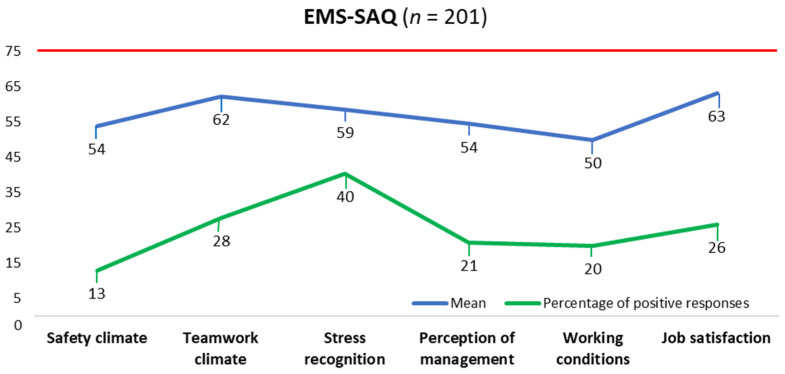
Overall mean scores and percentage of positive responses for each safety culture domain are based on triage nurses’ perceptions.

**Table 1 healthcare-11-03155-t001:** Participant characteristics.

Variables	*n*	%	*M* (*SD*)
Gender (*n* = 201)			-
Male	42	21	-
Female	159	79	-
Education (*n* = 201)			
Vocational nursing education	42	21	-
Bachelor’s degree in nursing	139	69	-
Master’s degree in nursing	20	10	-
Age (*n* = 201)			37 (9.89)
≤30	57	28	-
31–40	87	43	-
41–50	37	18	-
≥51	20	10	-
Working experience (*n* = 201)			14 (10.34)
≤5 years	53	26	-
6–10 years	38	19	-
11–15 years	45	22	-
≥16 years	65	32	-
Working experience in ED (*n* = 201)			9 (8.40)
≤5 years	99	49	-
6–10 years	46	23	-
11–15 years	24	12	-
≥16 years	32	16	-
MTS triage performance in years	201	100	5 (3.45)
Daily contact with patients	201	100	63 (40.81)
Duration of triage working day (*n* = 201)			-
4 h	2	1	-
8 h	104	52	-
12 h	95	47	-
No. of consecutive days in triage (*n* = 201)			-
One day	18	9	-
Two days	25	12	-
Three days	29	14	-
Four days	29	14	-
More than four days	100	50	-

Note: *n*—Sample size, *M*—Mean, *SD*—Standard deviation, ED—Emergency department, MTS—Manchester triage system.

**Table 2 healthcare-11-03155-t002:** Assessment of EMS-SAQ scores according to domains.

Domain	No. of Items	Min–Max	*M* (*SD*)	Mdn (IQR)
Total EMS-SAQ	30	14–85	57.27 (13.81)	56.67 (47.50–67.92)
Safety climate	7	11–96	53.82 (18.50)	53.57 (42.86–64.29)
Teamwork climate	6	13–100	62.31 (16.32)	62.50 (50–75)
Stress recognition	4	0–100	58.55 (24.20)	62.50 (37.50–75)
Perception of management	4	6–94	54.45 (21.02)	56.25 (37.50–68.75)
Working conditions	4	13–94	49.91 (17.37)	43.75 (37.50–62.50)
Job satisfaction	5	10–100	63.18 (17.19)	60 (50–75)

Note: *M* = Mean, *SD* = Standard deviation, Mdn = median, IQR = Interquartile range, Min–Max = Minimum–Maximum.

**Table 3 healthcare-11-03155-t003:** Differences in perceptions of positive and negative responses to safety culture according to respondents’ demographic characteristics.

Variables	<75 *n* (%)	>75 *n* (%)	U or *χ*^2^	df	*p* *
Gender					
Female	136 (86)	20 (17)	3316.50	1	0.907
Male	37 (88)	5 (12)			
Age					
≤30	49 (28)	8 (32)	17.750	3	<0.001 *
31–40	82 (47)	5 (20)			
41–50	33 (19)	4 (16)			
≥51	12 (7)	8 (32)			
Education					
Vocational NE	41 (98)	1 (2)	6.957	2	0.031 *
BSN	120 (86)	19 (14)			
MSN/MN	15 (75)	5 (25)			
Working experience					
≤5 years	45 (85)	8 (15)	8.875	3	0.031 *
6–10 years	33 (87)	5 (13)			
11–15 years	45 (100)	0 (0)			
≥16 years	53 (82)	12 (19)			

Note: BSN—Bachelor of Science in Nursing, MSN—Master of Science in Nursing, MN—master’s in nursing, NE—nursing education, *χ*^2^—Kruskal–Wallis/Chi-square, U—Mann–Whitney U test, *p*—*p*-value of the test, * = significant if <0.05, df = degrees of freedom.

## Data Availability

Additional data from this study are not publicly available in order to maintain participants’ anonymity but can be provided on request.

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
