# Peer review of "Perceptions of Patient Safety Culture among Triage Nurses in the Emergency Department: A Cross-Sectional Study"

_healthcare, 2023, doi:10.3390/healthcare11243155_

Round 1

Reviewer 1 Report

Comments and Suggestions for Authors

Dear authors,

Many thanks for submitting your valuable work to the journal. Please find attached my evaluation report, which discusses the issues that must be addressed for the improvement of your manuscript quality.

Many thanks

Best regards

Author Response

Responses to Reviewer 1

Reviewer 1 Comments

Methods

Comment 1: From my point of view, one of the main concerns of the present work is the inclusion in your sample of nursing assistants (21% of the study sample), along with registered nurses. Are nurses' assistants in charge of triage duties within the emergency department in the healthcare system of your country? I think that this is an erroneous practice because of the inadequate knowledge and scientific background of nurses' assistants. Alongside this, the inclusion of nursing assistants in your study can affect the reliability of the findings of the present study, leading to problematic findings regarding the patient safety culture of triage nurses. Additionally, the title of the present study is not representative because you refer to triage nurses and not to nursing personnel in general, in consideration of your study sample.

Response 1: Thank you for your comment and your concern. In Slovenia, triage nurses are licensed through specialisation at the Chamber of the Nurses' Association. The prerequisite for gaining this license is a bachelor's degree in nursing and at least three years of practical experience in emergency nursing. According to Slovenian legislation, triage can also be performed by expe-rienced (with experience working in an emergency department) nurse assistants who have also acquired specialised knowledge through the Slovenian Nursing Chamber and based on an employer's assessment that they are suitable to undertake triage duties., and we added it to the manuscript to be clearer to the readers.

Page/Line: 2/91-99

In Slovenia, triage is performed according to the Manchester Triage System (MTS) principles, which have proven to be the most suitable for implementation in our environment. Moreover, triage is performed by a registered nurse who has acquired additional specialised knowledge in this area and possesses a minimum of three years of experience in the emergency field. According to Slovenian legislation, triage can also be performed by experienced (with experience working in an emergency department) nurse assistants who have also acquired specialised knowledge through the Slovenian Nursing Chamber and based on an employer's assessment that they are suitable to undertake triage duties.

Comment 2: You state that the instrument used for data collection purposes was the Emergency Medical Services-Safety Attitudes Questionnaire (EMS-SAQ) validated tool (page 3, lines 105–106). I would like to clarify that this tool has been previously translated into the Slovenian language and validated among the Slovenian population, providing the relevant citation (validation study).

Response 2: Thank you for your comment. The validation of the Emergency Medical Services-Safety Attitudes Questionnaire (EMS-SAQ) was done according to Polit & Beck (2021), which also includes translation and psychometric testing, namely, internal consistency was assessed using Cronbach's ɑ coefficient and item-total correlations. Intraclass correlation coefficients were calculated to assess test-retest reliability. Exploratory factor analysis was also performed using the Kaiser-Meyer-Olkin index of sampling adequacy and the Bartlett test. The factor analysis was used to determine whether sufficient covariance was in the scale items. The validation paper of Slovenian validation of the Emergency Medical Services-Safety Attitudes Questionnaire (EMS-SAQ) itself cannot be referenced as it is still in the pre-review phase.

Comment 3: Another important concern is the reliability of the used tool, based on your statement that this instrument remains an underutilised tool in practice (page 2, line 65). Please, clarify.

Response 3: Thank you for your comment and clarification. After reassessing the sentence, it has been found that there has been a misinterpretation of its meaning. The questionnaire was an underutilised tool in prehospital care, not in the ED, as evidenced by the submitted individual studies. This error has been corrected in the paper.

Comment 4: Your data were collected during the COVID-19 pandemic period. Did this period affect the data collection process or create limitations regarding the assessment of the safety culture in the emergency department?

Response 4: Thank you for your comment. Indeed, the data were collected during the COVID-19 pandemic. The pandemic introduced unique challenges and circumstances that could impact assessing safety culture in the emergency department. We have this aspect added to the limitation section.

Page/Line: 10/353-363

Data were collected during the COVID-19 pandemic period. The pandemic introduced unique challenges and circumstances that could impact assessing safety culture in the emergency department. The limitations and effects on the data collection process during the COVID-19 pandemic may include disruptions in regular workflows, increased workload and stress among healthcare professionals, changes in protocols and procedures, and potential shifts in the focus of healthcare priorities. These factors could influence the responses of participants and the overall dynamics of the emergency department, potentially impacting the nuanced understanding of safety culture.

Comment 5: Page 6, 1st paragraph, lines 185–1187: You have given interpretations of your findings. However, the results section should present only the findings without comments or explanations. This task will be presented in the discussion section. Please correct.

Response 5: Thank you for your comment. We have rewritten this paragraph and removed indicated comments or explanations.

Comment 6: Page 7, last paragraph of the "Results" section, lines 218–220. Please give more details regarding the significant findings of the inferential statistics. For instance, you can state that a higher level of education is associated with significantly higher patient safety perceptions, etc.

Response 6: Thank you for your comment, and we have taken note. We have rewritten these findings.

Comment 7: "Discussion" section: This section is well written. However, I suggest giving a synopsis of the main findings of your study in the first paragraph. Additionally, in the last paragraph of this section (page 9, lines 300–327), you have effectively discussed the findings of the inferential statistics of your study. However, the significant association between age and perceived patient safety has been omitted. Please correct this.

Response 7: Thank you for your comment, and we have taken note. We have rewritten the first paragraph.

Page/Line: 7/231-241

This study aimed to assess the perception of patient safety culture among triage nurses in emergency departments. The study's findings reveal that the perceptions of triage nurses were negative across all analysed variables, underscoring the necessity for improvements in this domain. The aspect requiring the most significant improvement was working conditions. Conversely, job satisfaction had the highest average rating. The perception of positive and negative safety culture responses was statistically significant for age, education and length of working experience. The domain with the largest percentage of positive responses was stress recognition, followed by job satisfaction and teamwork climate. These findings highlight the need for improving safety culture across various dimensions of patient care within the triage process within the studied emergency departments.

Comment 8: In the study limitations, please include the underutilisation of the used instrument (EMS-SAQ).

Response 8: Thank you for the comment, and we have added this to the limitations.

Comment 9: Abstract, line 13: Please rewrite the sentence "In international countries, including Slovenia,…" as "In several countries, including Slovenia,…"

Response 9: Thank you for the comment, and we have rewritten the sentence.

Comment 10: Please add in the abstract the inferential statistics results.

Response 10: Thank you for the comment, and we have added the inferential statistics results in the abstract.

Comment 11: Page 2, line 76: Please rewrite it as follows: "…the primary objective of this study was to examine…"

Response 11: Thank you for the comment, and we have rewritten it as you suggested.

Comment 12: Page 2, line 78: Please rewrite as follows: "…we, also aimed to gain…"

Response 12: Thank you for the comment, and we have rewritten it as you suggested.

Comment 13: After the aim of the study, I suggest you state what new information your study intends to add to the existing body of literature.

Response 13: Thank you for the comment, and we have added this after the aim.

Page/line: 2/80-85.

By focusing on this unique context, the study intends to add valued insights into the perspectives of triage nurses, contributing valuable information to the existing body of literature on advanced patient safety practices in the emergency department.

Comment 14: Page 3, line 125: Please move the following sentence, "…, the data were entered into SPSS Statistic 25 software" to the "Data Analysis" section.

Response 14: Thank you for the comment, and we have moved the following sentence into the Data analysis section.

Comment 15: Please provide the findings regarding the Cronbach alpha coefficient of the EMS-SAQ (pages 3–4, lines 145–147) in the "Results" section.

Response 15: Thank you for the comment. We have moved the findings regarding the Cronbach alpha coefficient into the Results section.

Comment 16: Page 6, line 185: Please correct and write "(Table 2)" instead of (Figure 1).

Response 16: Thank you for the comment. We have corrected it.

Comment 17: Page 6, line 209: Please delete the word "n" that follows the word "domain'.

Response 17: Thank you for the comment, and we have rewritten this sentence.

Reviewer 2 Report

Comments and Suggestions for Authors

This is an interesting and well-written paper evaluating ED triage nurses’ perceptions of patient safety culture.  Measuring safety culture is important not only for patient outcomes, but also for staff wellbeing.  There are only few areas that require clarification, however an important detail in the results needs confirming.

Introduction: very well written, defines and explains patient safety succinctly, and justifies the study.

Materials and Methods:

2.2: a bit more information on the setting of the study (hospital locations, size, etc) would be helpful for non-local readers and would provide useful context.  Are the hospitals only in regional areas, i.e. outside metropolitan areas? 

2.3: please clarify if the survey was anonymous (there is a suggestion later that it was)

Results: 

Line 185: “….that triage nurses perceive the safety climate negatively…”.  Should “climate” be “culture”?

Lines 200-204: have the scores relating to items 49,48,and 33 (and possibly other negatively-worded items) been flipped as suggested in the methods?  For example, a low score for “Adverse events occur in the emergency department” should be preferable to a score rated above 70%.  From the wording of the results, it appears they have been included using the raw scores which would greatly impact the overall results.  Please check and confirm these results have been appropriately rated.

Line 209: Typo “….for each domain n of safety culture…”.

Discussion:

Comprehensive and concise.  Well written, great justification (dependent on clarification of results). 

Line 309: Alquwez et al (ref 31) – all authors listed which isn’t consistent with the rest of the paper.

Conclusion:

Good summary of the findings (dependent on results clarification). 

Author Response

Responses to Reviewer 2

Reviewer 2 Comments

Overall Comment: This is an interesting and well-written paper evaluating ED triage nurses’ perceptions of patient safety culture. Measuring safety culture is important not only for patient outcomes, but also for staff wellbeing. There are only few areas that require clarification, however an important detail in the results needs confirming.

Response to overall comment: Thank you very much for your comment.

Comment 1 (Introduction): very well written, defines and explains patient safety succinctly, and justifies the study.

Response 1: Thank you very much for your comment.

Comment 2 (Materials and Methods): 2.2: a bit more information on the setting of the study (hospital locations, size, etc) would be helpful for non-local readers and would provide useful context. Are the hospitals only in regional areas, i.e. outside metropolitan areas?

Response 2: Thank you for your comment; we have added more information to the manuscript. In Slovenia, hospitals are located in the cities and are accessible from all parts of the country from 30 to one hundred.

Comment 3 (Materials and Methods): 2.3: please clarify if the survey was anonymous (there is a suggestion later that it was)

Response 3: Thank you for your comment; we have added more information to the manuscript. In Slovenia, hospitals are located in the cities and accessible to all citizens from all parts of the country.

Comment 4 (Results): Line 185: “….that triage nurses perceive the safety climate negatively…”. Should “climate” be “culture”?

Response 4: Thank you for the comment; we have corrected it.

Comment 5 (Results): Lines 200-204: have the scores relating to items 49,48,and 33 (and possibly other negatively-worded items) been flipped as suggested in the methods? For example, a low score for “Adverse events occur in the emergency department” should be preferable to a score rated above 70%. From the wording of the results, it appears they have been included using the raw scores which would greatly impact the overall results. Please check and confirm these results have been appropriately rated.

Response 5: Thank you for your concern and comment. We have checked and confirmed that these results have been appropriately rated, as the authors of the questionnaire suggest.

Comment 6 (Results): Line 209: Typo “….for each domain n of safety culture…”.

Response 6: Thank you for the comment; we have corrected it.

Comment 7 (Discussion): Comprehensive and concise.  Well written, great justification (dependent on clarification of results). 

Response 7: Thank you for the comment.

Comment 8 (Discussion): Line 309: Alquwez et al (ref 31) – all authors listed which isn’t consistent with the rest of the paper.

Response 8: Thank you for the comment; we have corrected it.

Comment 9 (Conclusion): Good summary of the findings (dependent on results clarification).

Response 9: Thank you for the comment.

Reviewer 3 Report

Comments and Suggestions for Authors

This unroriginal study could be domewhat more attractive with strong conclusions. However, the conclusions are few, it is necesaary to complete this section.

Author Response

Responses to Reviewer 3

Reviewer 3 Comments

Comment 1: This unroriginal study could be domewhat more attractive with strong conclusions. However, the conclusions are few, it is necesaary to complete this section..

Response 1: Thank you for your comment. This study is original and interesting because it assesses how nurses in the triage ED perceive the patient safety culture. Measuring safety culture is important not only for patient outcomes but also for staff wellbeing. The study emphasises that triage nurses in emergency departments have a negative perception of safety culture, which requires immediate attention to improve various aspects of patient care in the triage process. This study is particularly noteworthy as one of the pioneering studies exploring triage nurses' perceptions of safety culture. By spotlighting this specific aspect, the study contributes to our understanding of the challenges frontline healthcare providers face and positions itself as a catalyst for further investigations in this critical domain.

Moreover, this study bridges an existing knowledge gap by elucidating the elements that triage nurses deem crucial in establishing a robust culture of safety in the emergency department. As such, it informs the academic discourse and provides practical insights to inform policy and practice to ensure patient safety during the triage process. We have also refined the conclusion to make it more attractive.

Round 2

Reviewer 1 Report

Comments and Suggestions for Authors

Dear authors,

Many thanks for your revision based on my previous comments on your work. In my opinion, the only remaining issue is the absence of validation in the Slovenian population of the EMS-SAQ, which significantly affects the reliability of the collected data. This validation study should have come before the present study.

Kind regards

Author Response

Responses to Reviewer 1

Comment 1: Many thanks for your revision based on my previous comments on your work. In my opinion, the only remaining issue is the absence of validation in the Slovenian population of the EMS-SAQ, which significantly affects the reliability of the collected data. This validation study should have come before the present study.

Response 1: Thank you for your comment. Regarding your concern about the lack of validation of the EMS-SAQ in the Slovenian population, we would like to clarify that the validation in the Slovenian population was performed before the start of this study. The validation results show that the questionnaire is suitable and reliable for use in the Slovenian environment. Therefore, we have used a reliable and credible questionnaire in this study. The validation document is in the pre-review stage for publication in a journal and, therefore, cannot be referenced.

Reviewer 2 Report

Comments and Suggestions for Authors

Thank you for addressing the feedback.  Your amendments have been incorporated well into the manuscript.  There are a few small points to be finalised, please:

Small typo line 81: "...we, also aimed to gain...." extra comma after "we"

Line 97: "...latest stats...." should be "latest statistics", and appropriately referenced

Line 101: please include reference for MTS in this section (no references in this additional paragraph)

Line 234: you state there is a “gender-based disparity” yet in the previous line you say there is a comparable trend in both males and females and haven’t noted anywhere a difference (Table 3 suggests there is no difference).  Please clarify this

A general comment:  for clarity, if you are referring to an EMS-SAQ domain name, please treat consistently (either italicised or as a proper noun, and use/not use quotation marks consistently).  There are a few areas where it is difficult to determine if you are referring to, for example, the domain “Working Conditions” or working conditions in general.

I am unsure why comments around the utilisation of the EMS-SAQ in the pre-hospital setting have been included, but assume it is in response to another reviewer’s feedback.  This has been well done despite the context of the study not being related to the pre-hospital setting.

Author Response

Responses to Reviewer 2

Reviewer 2 Comments

Overall Comment: Thank you for addressing the feedback. Your amendments have been incorporated well into the manuscript. There are a few small points to be finalised, please.

Response to overall comment: Thank you very much for your comment.

Comment 1: Small typo line 81: "...we, also aimed to gain...." extra comma after "we"

Response 1: Thank you for your comment. We have corrected it.

Comment 2: Line 97: "...latest stats...." should be "latest statistics", and appropriately referenced.

Response 2: Thank you for your comment. We have corrected it and have added the appropriate reference.

Comment 3: Line 101: please include reference for MTS in this section (no references in this additional paragraph)

Response 3: Thank you for your comment. We have referenced this.

Comment 4: Line 234: you state there is a "gender-based disparity" yet in the previous line you say there is a comparable trend in both males and females and haven't noted anywhere a difference (Table 3 suggests there is no difference). Please clarify this

Response 4: Thank you for the comment. We apologise for the mistake, and you are right. There was no difference between genders, and we have corrected it.

Comment 5: A general comment: for clarity, if you are referring to an EMS-SAQ domain name, please treat consistently (either italicised or as a proper noun, and use/not use quotation marks consistently). There are a few areas where it is difficult to determine if you are referring to, for example, the domain "Working Conditions" or working conditions in general.

Response 5: Thank you for your comment. We have corrected it.

Comment 6: I am unsure why comments around the utilisation of the EMS-SAQ in the pre-hospital setting have been included, but assume it is in response to another reviewer's feedback. This has been well done despite the context of the study not being related to the pre-hospital setting.

Response 6: Thank you for your comment. We agree with you, and we have included it in the previous revision at the request of another reviewer.
